# Association of Anti-EGFR Antibody and MEK Inhibitor in Gynecological Cancer Harboring RAS Mutation: A Case Series

**DOI:** 10.3390/ijms23063343

**Published:** 2022-03-19

**Authors:** Julie Niogret, Valentin Derangère, Corentin Richard, Lisa Nuttin, François Ghiringhelli, Laure Favier, Leila Bengrine Lefevre, Anthony Bergeron, Laurent Arnould, Romain Boidot

**Affiliations:** 1Department of Medical Oncology, Georges-François Leclerc Cancer Center, 21000 Dijon, France; jniogret@cgfl.fr (J.N.); fghiringhelli@cgfl.fr (F.G.); lfavier@cgfl.fr (L.F.); lbengrine@cgfl.fr (L.B.L.); 2Platform of Transfer in Biological Oncology, Georges-François Leclerc Cancer Center, 21000 Dijon, France; vderangere@cgfl.fr (V.D.); lnuttin@cgfl.fr (L.N.); 3Unit of Molecular Biology, Department of Biology and Pathology of Tumors, Georges-François Leclerc Cancer Center, 1 Rue du Professeur Marion, CEDEX, 21079 Dijon, France; crichard@cgfl.fr; 4Genetic and Immunology Medical Institute (GIMI), 21000 Dijon, France; 5Unit of Pathology, Department of Biology and Pathology of Tumors, Georges-François Leclerc Cancer Center, 21000 Dijon, France; abergeron@cgfl.fr (A.B.); larnould@cgfl.fr (L.A.); 6CNRS Unit U6302, 21000 Dijon, France

**Keywords:** low-grade serous ovarian carcinoma, *RAS* mutation, MEK inhibitors

## Abstract

Low-grade serous carcinoma represents a minority of serous carcinoma. Although they have better prognosis than high-grade serous carcinoma, they respond poorly to chemotherapy. Thus, it appears necessary to find other treatments such as targeted therapies. Since *RAS* or *RAF* mutations occur frequently in low-grade serous carcinoma and lead to constitutively activated MAPK cascade, MEK inhibition should be effective in the treatment of low-grade serous carcinoma. So, we wanted to evaluate the clinical benefit of MEK inhibitors in the management of advanced-stage low-grade serous carcinoma harboring *KRAS* or *NRAS* mutation. We report a case series of three women with advanced-stage low-grade serous carcinoma harboring *RAS* mutation who had stabilization of their disease during several months under targeted therapy combining anti-EGFR antibody and MEK inhibitor. We performed in vitro experiments, confirming the effectiveness of MEK inhibitor on the *KRAS*-mutated OVCAR-5 cell line, and the constitutively activation of MAPK cascade in *RAS*-mutated carcinoma. However, it seems that the anti-EGFR antibody does not provide any additional benefit. After whole exome analysis is carried out on the patient with the shortest response, we observed the appearance of RB1 loss-of-function mutation that could be a mechanism of resistance to MEK inhibitors in *RAS*- of *RAF*-mutated cancers. The MEK inhibitor is effective in the advanced stages of low-grade serous carcinoma harboring *RAS* mutation with acceptable tolerance. RB1 loss could be a mechanism of resistance to MEK inhibitors in *RAS*-mutated low-grade serous carcinoma.

## 1. Introduction

Ovarian carcinoma is the leading cause of death from gynecologic malignancies in developed countries [1]. Low-grade serous ovarian carcinoma represents a minority of ovarian serous carcinoma—about 10% of all serous ovarian carcinoma [2]. Women with low-grade serous ovarian carcinoma are diagnosed at a younger age and have a longer overall survival than women with high-grade serous ovarian carcinoma. Furthermore, low-grade serous ovarian carcinoma is less aggressive than high-grade serous ovarian carcinoma. Although women with low-grade serous ovarian carcinoma usually have an indolent clinical history, they have multiple recurrences and may die of this disease [3].

Although the overall prognosis is better in women with low-grade than that in high-grade serous ovarian carcinoma, women with low-grade serous ovarian carcinoma have lower response rates to conventional ovarian carcinoma treatments. Since low-grade serous ovarian carcinomas are less responsive to conventional chemotherapy than high-grade serous ovarian carcinomas, it appears necessary to find other treatments, such as targeted therapies [2]. Effective and high-quality evidence-based treatment options for advanced-stage, low-grade serous ovarian carcinoma are lacking.

The classic mitogen-activated protein kinase (MAPK) cascade, also called the RAS/RAF/MEK/ERK pathway, is one of the major biologic pathways frequently altered in human carcinoma [4], mainly by constitutive activation of RAS and RAF proteins [5]. *BRAF*, *KRAS,* and *NRAS* mutations occurred in approximately 33%, 35%, and 20%, respectively, of all low-grade serous ovarian carcinoma [6,7]. *BRAF* and *KRAS* mutations occurred in approximately 2% and 19%, respectively, of advanced-stage low-grade serous ovarian carcinoma [8]. While mutations in *RAS* or *RAF* can lead to constitutively activated extracellular signal-regulated kinase (ERK), inhibition of MEK theoretically leads to specific blockade of ERK due to the lack of redundancy of that portion of the cascade [5].

We wanted to evaluate the clinical benefit of MEK inhibitors in the management of advanced-stage low-grade serous carcinoma harboring *KRAS* or *NRAS* mutation. Here, we present a case series of women with advanced-stage, low-grade serous carcinoma harboring *KRAS* or *NRAS* mutation treated with targeted therapy combining anti-EGFR antibody and MEK inhibitor. Then, we present in vitro experiments to determine the contribution of each of these treatments.

## 2. Presentation of Cases

### 2.1. Methods and Materials

This study on patient samples was conducted in accordance with the Declaration of Helsinki and approved by the Ethics Committee of the Georges-François Leclerc Cancer Center (Dijon, France) and by the Consultative Committee of Burgundy (Dijon, France) for the Protection of Persons Participating in Biomedical Research (Comité Consultatif de Protection des Personnes en Recherche Biomédicale de Bourgogne). Written informed consent was provided.

#### 2.1.1. Exome Sequencing

The DNA was extracted using the Maxwell-16 FFPE Plus LEV DNA purification kit (Promega Corporation, Madison, WI, USA) according to the manufacturer’s protocol. The DNA quality was assessed by spectrophotometry with absorbance measured at 230, 260 and 280 nm. The DNA was quantified by a Qubit fluorometric assay (cat. no. Q32850; Thermo Fisher Scientific Inc., Waltham, MA, USA), according to the manufacturer’s instructions.

Libraries were constructed from 200 ng DNA and captured using the SureSelect Human All Exon v6 kit (Agilent, Santa Clara, CA, USA) following the manufacturer’s protocol. Paired-end (2 × 111 bases) sequencing was performed on a NextSeq500 device (Illumina Inc., San Diego, CA, USA). Next, the sequences were aligned and annotated with the human Hg19 genome based on the SureSelect Human all Exon v6 manifest using the BWA and GATK algorithms. Only sequences with a read depth of 10×, a mutation allele frequency >5%, and a frequency in the general population inferior to 1% were retained for further analysis.

#### 2.1.2. Cell Culture

The OVCAR-5/RFP cell line (Cell Biolabs, San Diego, CA, USA) (ovarian cancer cells harboring *KRAS* p.G12V mutation) was cultured using the culture medium recommended by the manufacturer. They were then put on a 96-well flat bottom plate at the concentration of 200 cells in 200 µL of culture medium in each well. After 24 h in an incubator at 37 °C, the culture medium was removed and replaced by 200 µL of culture medium containing increasing doses of trametinib (MEK inhibitor, 0 to 100 nM) or cetuximab (monoclonal antibody anti-EGFR, 0 to 15 µM) or both. After 5 days of treatment in an incubator at 37 °C, we evaluated cellular viability by crystal violet coloration. We performed these experiments three times with three replicates each time.

#### 2.1.3. Immunohistochemistry

FFPE samples were sliced with a thickness of 4 µm using a microtome. After dewaxing and heat-induced epitope retrieval (TRS High pH, Agilent K8000 kit, Agilent, Santa Clara, CA, USA) during 50 min at 95 °C in an Agilent PT Link module, samples were washed with wash buffer (Agilent K8000 kit, Agilent, Santa Clara, CA, USA) two times for 5 min. Then samples were incubated within the autostainer apparatus from Agilent with peroxidase blocking reagent (Agilent K8000 kit, Agilent, Santa Clara, CA, USA) for 5 min. Anti-Phospho-Erk antibody (CST #4370) was then applied at RT during 20 min at a final concentration of 1/400 in Ab diluent (Enzo ADI-950-244-0250, Enzo Life Sciences, Farmingdale, NY, USA). After washing, HRP-conjugated polymers (Agilent K8000 kit, Agilent, Santa Clara, CA, USA) were then incubated during 10 min at RT. A freshly reconstituted DAB solution (Agilent K8000 kit, Agilent, Santa Clara, CA, USA) was then applied for 10 min to develop signal. After washing, hematoxylin (Enzo HighDef ENZ-ACC106-0100, Enzo Life Sciences, Farmingdale, NY, USA) was incubated for 20 min to counterstain nuclei. Slides were finally washed two times using DI water and then permanently mounted according to the pathology laboratory protocol. Slides were digitized with Hamamatsu HT2.0 slide scanner.

#### 2.1.4. Data Availability

Genomic data could be shared upon reasonable request to the corresponding author in accordance to French law for genomic data.

### 2.2. Results

We report cases of three women with recurrent advanced-stage, low-grade serous carcinoma. All three benefited from a whole exome analysis by next-generation sequencing of the tumor, which objectified *KRAS* or *NRAS* mutation. Because of these mutations, the molecular tumor board recommended treating these patients with an association of anti-EGFR antibody and MEK inhibitor, as accomplished for *KRAS*-mutated colon carcinoma in our cancer center.

The first case is a 43-year-old woman with low-grade stage IV serous ovarian carcinoma harboring a *NRAS* mutation in exon 3 p.Gln61Arg (p.Q61R). At diagnosis, she had skin metastasis. In the first line of treatment, she underwent surgery (total hysterectomy with bilateral adnexectomy, lombo-aortic lymphadenectomy, omentectomy, and appendicectomy) and then received six cures of cisplatin plus gemcitabin which allowed complete response. She progressed 14 months after surgery and received six cures of trabectidin plus pegylated liposomal doxorubicin as the second line treatment as the disease stabilized. She progressed for 7 months after starting the second line of treatment and received 6 cures of carboplatin plus gemcitabin plus bevacizumab in the third line of treatment, followed by 26 cures of Bevacizumab as maintenance treatment, which stabilizes the disease. She progressed under maintenance treatment with bevacizumab after 22 months and received a fourth line of treatment with carboplatin plus pegylated liposomal doxorubicin for six cures as the disease stabilized. She progressed for 5 months after starting the fourth line of treatment and received 2 months of tamoxifen after the fifth line of treatment as the disease progressed. After exome analysis, she was treated by panitumumab (monoclonal antibody anti-EGFR) and cobimetinib (MEK inhibitor) in the sixth line of treatment. At that time, she had skin and peritoneal metastasis. She received 10 cures of the association. This association allowed stabilization of the disease (Figure 1A) with a progression-free survival (PFS) of 4.4 months, and a grade 2 dermatological toxicity according to CTCAE4.0. After recurrence, she benefited from a new biopsy with whole exome analysis by next-generation sequencing.

The second case is a 36 years old woman with mesonephric-like carcinoma of the endometrium harboring a *KRAS* mutation in exon 2 p.Gly12Asp (p.G12D). At diagnosis, she had peritoneal metastasis. In first line, she underwent surgery (total hysterectomy with bilateral adnexectomy, lombo-aortic lymphadenectomy and omentectomy), received 6 cycles of Cisplatine plus Paclitaxel and then benefited from pelvic radiotherapy and vaginal brachytherapy which allowed complete response. She progressed 19 months after surgery and received 9 cycles of Carboplatin plus Paclitaxel as second line with complete response. She progressed 9 months after starting second line and received in third line 6 cures of pegylated liposomal Doxorubicin with progressive disease. She was treated by panitumumab and cobimetinib in the fourth line of treatment. At that time, she had liver and peritoneal metastasis. She received 17 cures of the association. This association allowed partial response (Figure 1B) with a PFS of 9.1 months, and a grade 2 dermatological toxicity according to CTCAE4.0.

The third case is a 44 years old woman with low-grade stage IV serous ovarian carcinoma harboring a *KRAS* mutation in exon 2 p.Gly12Val (p.G12V). She had a history of borderline serous cystadenomas with endocystic development in both ovaries treated by left then right adnexectomy. At diagnosis, she had endometrial, appendicular, pleural, peritoneal and bones metastasis. In first line, she underwent surgery (one-piece hysterectomy and rectosigmoidectomy with lombo-aortic and bilateral ilio-pelvic lymphadenectomy, omentectomy, appendicectomy and removal of the parietal peritoneum of the right diaphragmatic dome) and then received 6 cycles of Cisplatine plus Paclitaxel which allowed complete response. She progressed 35 months after surgery and received in second line 1 cycle of Carboplatin plus Paclitaxel then 14 cycles of Paclitaxel with stabilization of the disease. Carboplatin was stopped because of anaphylaxis. She progressed 46 months after starting second line and received in third line 1 cycle of Cisplatin plus Paclitaxel then 5 cycles of Cisplatin with stabilization of the disease. Paclitaxel was stopped because of anaphylaxis. She progressed 17 months after starting third line and received in fourth line 5 months of Tamoxifen with stabilization of the disease. She was treated by panitumumab and cobimetinib in the fifth line of treatment. At that time, she had endometrial, appendicular, pleural, peritoneal, bones, skin, and pulmonary metastasis. This association allowed partial response (Figure 1C), with a grade 2 dermatological toxicity, grade 1 diarrhea and grade 3 mucositis according to CTCAE4.0. She is currently still under treatment since 36.5 months and has received 66 cures.

#### 2.2.1. In Vitro Experiments

Firstly, we wanted to determine the contribution of MEK inhibitors and anti-EGFR antibody about the impact observed on patients. We treated OVCAR-5 cell line (ovarian cancer cells harboring *KRAS* p.G12V mutation) with increasing dose of Trametinib (Figure 2A) or Cetuximab (Figure 2B) or both and evaluated cellular viability by crystal violet coloration. Trametinib inhibited OVCAR-5 cell viability, but we did not observe any impact of Cetuximab alone or in combination with Trametinib on cell viability (Figure 2C). 

Secondly, since the phosphorylation of ERK is the result of the activation of the RAS/RAF/MEK/ERK pathway, we analyzed the expression of phosphorylated ERK (pERK) by immunohistochemistry using an antibody anti-pERK (Figure 3). Moreover, with colon cancer as a the reference in *RAS*/*RAF*-mutated carcinoma, we wanted to compare colon and serous carcinoma. We did not observe any expression of pERK in colon or serous *KRAS*/*NRAS*/*BRAF* wild-type carcinoma (Figure 3). Interestingly, we observed high pERK expression in serous carcinoma with *KRAS* p.G12V (tissue from carcinoma of case n°3) or *NRAS* p.Q61R (tissue from carcinoma of case n°1) mutation, but only low pERK expression in colon carcinoma with *KRAS* p.G12D mutation (Figure 3). Unfortunately, we had no more tumor tissue available for analyzing the expression of pERK in the case of the patient n°2.

#### 2.2.2. Clinical Observations

Firstly, this therapeutic association was modeled on that applied in mutated *KRAS* colon carcinoma in our anticancer center at that time. *RAS* mutations are predictive of the lack of efficacy of anti-EGFR antibody in patients with metastatic colon carcinoma. Queralt et al. observed, in vitro, a benefit of the association of anti-EGFR antibody and MEK inhibitor on *NRAS*-mutant colorectal cancer cell viability [9]. Because of these promising pre-clinical evidence for synthetic lethality of this association in *NRAS* mutant colorectal cancer, we decided to treat patients with metastatic and chemo-resistant *KRAS* mutated colon carcinoma with the association of anti-EGFR antibody and MEK inhibitor. In our institution, eight patients with advanced-stage colon carcinoma (six with mutated *KRAS* and two *KRAS* wild type) were treated with the association of anti-EGFR antibody and MEK inhibitor. Interestingly, the median PFS is significantly higher (Log-rank *p*-value = 0.036) in *RAS* mutated serous carcinoma (n = 3, 274 days) than in *KRAS* mutated colon carcinoma (n = 6, 61 days). While there is no statistically significant difference (Log-rank *p*-value = 0.212) in PFS between *KRAS* mutated (n = 6, 61 days) and *KRAS* wild type (n = 2, 105 days) colon carcinoma (Figure 4). Therefore, we believe that this therapeutic association is effective in the management of advanced-stage of low-grade serous carcinoma harboring *RAS* mutation, but its interest in the management of advanced-stage colon carcinoma harboring *KRAS* mutation seems limited.

Secondly, we wanted to know what proportion of patients with serous carcinoma had *RAS*/*RAF* mutations. In our anticancer center, 129 patients with serous carcinoma benefited of whole exome analysis by next-generation sequencing of the tumor. Among them, eighty-six (66.7%) had high-grade serous carcinoma, thirteen (10.1%) had low-grade serous carcinoma, and thirty (23.3%) had serous carcinoma with unknown grade. Only one patient (1.2%) with high-grade serous carcinoma had *RAS*/*RAF* mutations, while eight patients (61.5%) with low-grade serous carcinoma had *RAS*/*RAF* mutations (Table 1). No patients had both *RAS* and *RAF* mutations, and no patients had multiple *RAS* or *RAF* mutations. Mutations are described in Table 1.

Finally, we wanted to know why the patient in the first case had a short response to the combination Panitumumab plus Cobimetinib. After recurrence, she benefited of a new biopsy of progressing tumor with whole exome analysis by next-generation sequencing. Comparing the two whole exome analysis we observed only the appearance of DNMT3A p.(Val227Glu) and RB1 p.(Arg696*) mutations.

## 3. Discussion

In our case series of 3 patients with a low-grade serous carcinoma, we observed that a treatment based on the association anti-EGFR antibody and MEK inhibitor tested in *RAS* mutated colon carcinomas was very efficient on *RAS* mutated serous carcinomas with a patient under treatment for more than 1000 days.

We observed that Trametinib inhibited OVCAR-5 cell viability, which is consistent with the antiproliferative effects of Trametinib observed on the same cell line by Campos et al. [10]. Moreover, in our in vitro experiments, Trametinib impact cell viability of ovarian *KRAS* mutated carcinoma, but Cetuximab does not provide any additional benefit. Thus, we believe that monotherapy with a MEK inhibitor would be as effective as and probably less toxic than the combination. We also observed that the RAS/RAF/MEK/ERK pathway seems to be more activated in serous *KRAS*/*NRAS* mutated carcinoma than in colon *KRAS* mutated carcinoma. This could explain the clinical observation of the better efficacy of MEK inhibitor in serous *KRAS*/*NRAS* mutated carcinoma.

Selumetinib, a MEK inhibitor, has been tested in low-grade serous carcinoma in a phase II clinical trial [11]. They include fifty-two patients. Selumetinib was active in the treatment of recurrent low-grade serous carcinoma with a median PFS of 11 months. Most patients had stable disease (65.4%), but there was also partial response (13.5%) and only one complete response (1.9%). Selumetinib was well tolerated. Its main toxicities were gastrointestinal and dermatologic toxicities. The data observed in our cases were consistent with that of this trial on safety and efficacy. Moreover, the results of this clinical trial are consistent with our in vitro results: monotherapy with a MEK inhibitor is effective. Surprisingly, in this study, response to Selumetinib did not appear to be related to *RAS*/*RAF* mutational status, while for other authors [12] *RAS* mutational status impacts the effectiveness of MEK inhibitors. Currently, an international randomized phase II/III clinical trial using Trametinib is ongoing (NCT02101788) and a translational research component to better understand the molecular mechanisms of MEK inhibitor efficacy is included. Preliminary results of the clinical trial NCT02101788, comparing Trametinib to physician’s choice standard of care in recurrent low-grade serous ovarian carcinoma, were presented at ESMO 2019 [13]. Trametinib was associated with significantly improved PFS (13 months compared to 7.2 months in control group, HR 0.48; 95% CI, 0.36–0.64; *p* < 0.0001) and objective response rate (26.2% compared to 6.2% in control group, OR 5.4; 95% CI, 2.39–12.21; *p* < 0.0001) [13]. Although the final results of this study are not available, Trametinib may be a new standard of care for recurrent low-grade serous ovarian carcinoma.

The *RB1* R696X mutation is a loss of function mutation. *RB1* (Retinoblastoma 1) is a tumor suppressor gene, which is altered in 4.03% of all cancers and 2.81% of ovarian carcinoma [14]. Loss of RB1 is thought to be involved in resistance to MEK inhibitors in mutated *BRAF* melanoma and mutated *KRAS* lung cancer cell lines [15,16]. Thus, we believe that RB1 loss may be a mechanism of resistance to MEK inhibitors in *RAS* or *RAF* mutated cancers.

## 4. Conclusions

We believe that monotherapy with a MEK inhibitor may have clinical benefit for women with advanced-stage low-grade serous carcinoma harboring *RAS* or *RAS* mutation, while being less toxic than combotherapy with anti-EGFR antibody and, we also believe that RB1 loss could be a mechanism of resistance to MEK inhibitors in *RAS* or *RAF* mutated cancers.

## Figures and Tables

**Figure 1 ijms-23-03343-f001:**
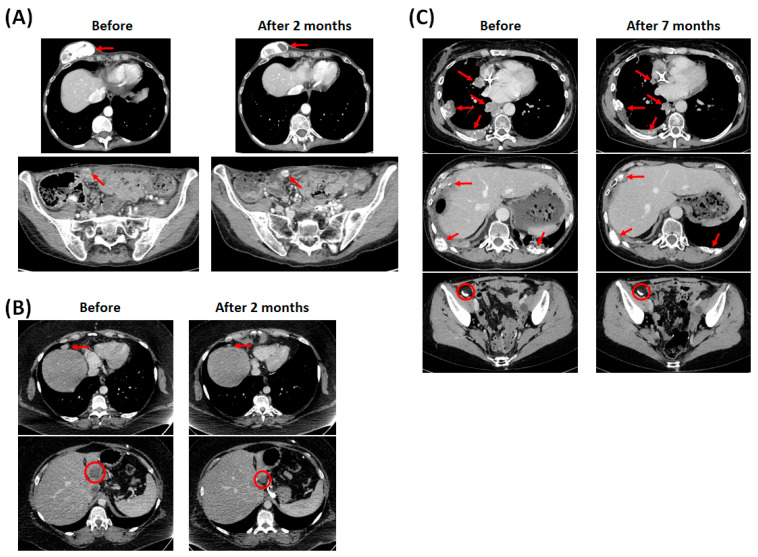
Computed tomography imaging results before and during the treatment of the anti-EGFR antibody and MEK inhibitor combination from patient of case n°1 (**A**), case n°2 (**B**) and case n°3 (**C**).

**Figure 2 ijms-23-03343-f002:**
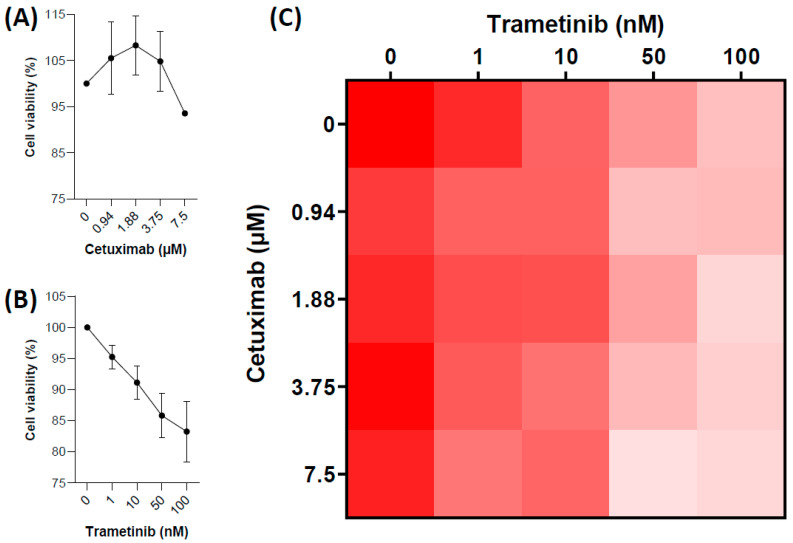
Effects of Trametinib or/and Cetuximab on human OVCAR-5 ovarian cancer cell line. Human OVCAR-5 ovarian cancer cell line was treated with indicated concentrations of Trametinib or/and Cetuximab for 48 h. (**A**,**B**). Cell viability was analyzed using crystal violet staining. (**A**,**B**). Relative absorbance compared to untreated according to the dose of Trametinib (**A**) or Cetuximab (**B**). (**C**) Relative viability of treatment combination. Experiments were performed three times with three replicates each time.

**Figure 3 ijms-23-03343-f003:**
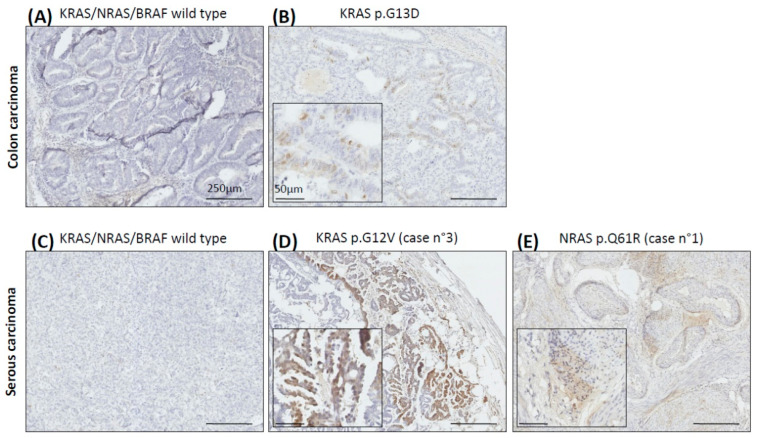
pERK expression in serous and colon carcinoma harboring or not *RAS/RAF* mutations. Serous and colon carcinoma biopsies from patients were fixed, embedded in paraffin and stained with anti-pERK. (**A**,**B**) shows pERK expression in colon carcinoma *RAS/RAF* WT (**A**) or *KRAS* mutated (**B**). (**C**–**E**) shows pERK expression in serous carcinoma *RAS/RAF* WT (**C**), *KRAS* mutated (**D**) or *NRAS* mutated (**E**). (**D**,**E**) are obtained from biopsy of case n°3 and n°1 respectively.

**Figure 4 ijms-23-03343-f004:**
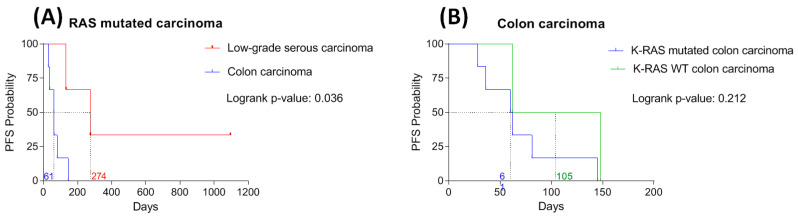
Association between histology and/or *RAS* mutational status and progression-free survival (PFS). (**A**) Kaplan-Meier survival curves for PFS according to histology in *RAS* mutated carcinoma. (**B**) Kaplan-Meier survival curves for PFS according to *RAS* mutational status in colon carcinoma.

**Table 1 ijms-23-03343-t001:** RAS/RAF mutations observed with whole exome analysis by next-generation sequencing of serous carcinomas in our cancer center.

		Tumor Grade		
Mutations	Total	High	Low	Unknown
	129 (100)	86 (66.7%)	13 (10.1%)	30 (23.3%)
*BRAF*	4 (3.1%)	0	2 (15.4%)	2 (6.7%)
V600E	3		2	1
D595N	1			1
*CRAF* (*RAF1*)	1 (0.8%)	0	0	1 (3.3%)
R143W	1			1
*KRAS*	7 (5.4%)	1 (1.2%)	5 (38.5%)	1 (3.3%)
G12C	2	1	1	
G12D	2		2	
G12V	2		2	
D119Y	1			1
*NRAS*	1 (0.8%)	0	1 (7.7%)	0
Q61R	1		1	

## Data Availability

Genomic data could be shared upon reasonable request to the corresponding author in accordance to French law for genomic data.

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
