# Peer review of "Association of Anti-EGFR Antibody and MEK Inhibitor in Gynecological Cancer Harboring RAS Mutation: A Case Series"

_ijms, 2022, doi:10.3390/ijms23063343_

Round 1

Reviewer 1 Report

Niogret et al. investigated the connection of anti-EGFR antibody and MEK inhibitor therapy with gynecological cancers harbouring RAS mutations in the current case study. The authors presented a case series of three women with advanced-stage low-grade serous carcinoma harbouring a Ras mutation who achieved disease stability over many months with a combination of anti-EGFR antibody and MEK inhibitor treatment. Additionally, whole-exome analysis in the patient with the quickest response revealed that the emergence of an RB1 loss-of-function mutation may be a cause of MEK inhibitor resistance in RAS or RAF mutant tumours. Finally, it was determined that MEK inhibitors are successful in advanced-stage low-grade serous carcinomas containing RAS mutations with tolerable tolerance, but RB1 deletion may be a mechanism of MEK inhibitor resistance in low-grade serous carcinomas harbouring RAS mutations. The study is beneficial and has several clinical diagnostic and therapeutic implications. However, some revisions and clarifications are necessary before publishing.

The third paragraph of the introduction is unnecessary and should be omitted; alternatively, authors might utilize the space to include specific information. My idea would be to provide information about the usage of additional downstream inhibitors for the alternative therapy option. In the end, however, writers are free to report and fill in the gaps as they see fit, particularly if they want to utilize the area to report on topics that go beyond academic expertise.

I cannot see a direct relationship between the in vitro experiment and the comprehensive clinical case report. My advice is to strengthen the relationship and explain why in vitro experiments are beneficial.

When it comes to protein selection, why just RB1? Numerous additional proteins downstream are also critical for the resistance of RAS mutant cancer types. The survival plot of RB1 in comparison to other downstream protein partners may be used to demonstrate the concept.

Author Response

Reviewer 1

Niogret et al. investigated the connection of anti-EGFR antibody and MEK inhibitor therapy with gynecological cancers harbouring RAS mutations in the current case study. The authors presented a case series of three women with advanced-stage low-grade serous carcinoma harbouring a Ras mutation who achieved disease stability over many months with a combination of anti-EGFR antibody and MEK inhibitor treatment. Additionally, whole-exome analysis in the patient with the quickest response revealed that the emergence of an RB1 loss-of-function mutation may be a cause of MEK inhibitor resistance in RAS or RAF mutant tumours. Finally, it was determined that MEK inhibitors are successful in advanced-stage low-grade serous carcinomas containing RAS mutations with tolerable tolerance, but RB1 deletion may be a mechanism of MEK inhibitor resistance in low-grade serous carcinomas harbouring RAS mutations. The study is beneficial and has several clinical diagnostic and therapeutic implications. However, some revisions and clarifications are necessary before publishing.

We thank the Reviewer for her/his remarks and comments. We hope that the revised version of the manuscript will meet her/his requirements for an acceptance. 

The third paragraph of the introduction is unnecessary and should be omitted; alternatively, authors might utilize the space to include specific information. My idea would be to provide information about the usage of additional downstream inhibitors for the alternative therapy option. In the end, however, writers are free to report and fill in the gaps as they see fit, particularly if they want to utilize the area to report on topics that go beyond academic expertise.

As proposed by the Reviewer, we deleted 2 sentences in the 3rd paragraph. Indeed, in agreement with the Reviewer, we found that these sentences were not necessary for the understanding of the study.  

I cannot see a direct relationship between the in vitro experiment and the comprehensive clinical case report. My advice is to strengthen the relationship and explain why in vitro experiments are beneficial.

In vitro experiments enabled us to show that the main part of the effect observed in patients was due to MEK inhibitor. Indeed, in vitro, anti-EGFR antibodies did not seem to have a significant effect on ovary cancer cell lines. To the contrary, MEK inhibitors use dramatically decreased the growth of KRAS mutated ovary cancer cells. The addition of anti-EGFR antibody did not increase the effect of MEK inhibitors. We added a sentence at the beginning of the In vitro experiments paragraph.   

When it comes to protein selection, why just RB1? Numerous additional proteins downstream are also critical for the resistance of RAS mutant cancer types. The survival plot of RB1 in comparison to other downstream protein partners may be used to demonstrate the concept.

We apologize if the Reviewer did not understand the RB1 Part. In fact, as the first patient had relapsed quickly, the physician decided to take a new biopsy of the progressive lesion. On these new lesions, whole exome sequencing was performed and, by comparing the new exome with the first one (performed on the diagnosis biopsy), we observed the appearance of 2 mutations: p.(Val227Glu) on DNMT3A and p.(Arg696*) on RB1. It is known that loss-of-function mutations on RB1 could be involved in resistance to treatment targeting EGFR pathway.

Reviewer 2 Report

Reviewer's report

To the authors:

In the paper: “Association of anti-EGFR antibody and MEK inhibitor in gynecological cancer harboring RAS mutation: a case series”, the authors aimed to study the effects of combined therapy anti-EGFR antibody and MEK inhibitor. They found that MEK inhibitor is effective in advanced stage of low-grade serous carcinoma harboring RAS mutation, however they concluded that anti-EGFR antibody does not provide any additional benefit. The study performed is relevant in the field what might have important clinical implications, however the manuscript needs several revisions as detailed above.

Revisions required:

  • Please revise all over the manuscript for the RAS symbol, gene symbol should be all caps in italic (RAS) and protein symbol should be all caps not italic (RAS).
  • The abstract needs to be improved. The authors should clearly emphasize the main aim of the study.
  • The keywords are missing, thus the authors should complete this section.
  • In the introduction section the authors should highlighting the aim of this study before describing what was performed.
  • The materials and methods section could begin by the description of the clinical cases used in the study, namely, the number of patients and their characterization by mutations as presented at section “2.2.2. Clinical observations” and table 1.
  • In the results, the authors should describe the recurrences (progression) always in the same scale (or months, or days). Regarding the in vitro experiments, it is missing the number of replicates and the statistical analysis, that should be included in the results and in the figure 2 legend. Additionally, the authors should better clarify why colorectal cancer patients samples were used. Moreover, the analysis of  DNMT3A V227E and RB1 R696X mutations should be included in the results section.
  • The conclusion needs to be rewritten, in order to highlight the main aim of the study, the results obtained and the clinical relevance of performing this study.

Author Response

Reviewer 2

Reviewer's report

To the authors:

In the paper: “Association of anti-EGFR antibody and MEK inhibitor in gynecological cancer harboring RAS mutation: a case series”, the authors aimed to study the effects of combined therapy anti-EGFR antibody and MEK inhibitor. They found that MEK inhibitor is effective in advanced stage of low-grade serous carcinoma harboring RAS mutation, however they concluded that anti-EGFR antibody does not provide any additional benefit. The study performed is relevant in the field what might have important clinical implications, however the manuscript needs several revisions as detailed above.

We thank the Reviewer for her/his comments and remarks. We hope that the revised version will meet her/his requirements for an acceptance of our work.  

Revisions required:

  • Please revise all over the manuscript for the RAS symbol, gene symbol should be all caps in italic (RAS) and protein symbol should be all caps not italic (RAS).

We apologize for this mistake. Throughout the manuscript, we put in italic gene symbol. We also modified the Figure 3.

  • The abstract needs to be improved. The authors should clearly emphasize the main aim of the study.

We added a sentence in the abstract that clearly indicates the goal of the study.

  • The keywords are missing, thus the authors should complete this section.

We apologize for this oversight. We added the keywords.  

  • In the introduction section the authors should highlighting the aim of this study before describing what was performed.

We added sentences to highlight the aim of the study.

  • The materials and methods section could begin by the description of the clinical cases used in the study, namely, the number of patients and their characterization by mutations as presented at section “2.2.2. Clinical observations” and table 1.

The study is a Case Report under the form of a Case series. In this kind of article, clinical observations are part of Results. Consequently, we decided to keep clinical cases in the Results section.   

  • In the results, the authors should describe the recurrences (progression) always in the same scale (or months, or days). Regarding the in vitro experiments, it is missing the number of replicates and the statistical analysis, that should be included in the results and in the figure 2 legend. Additionally, the authors should better clarify why colorectal cancer patients samples were used. Moreover, the analysis of  DNMT3A V227E and RB1 R696X mutations should be included in the results section.

We agree with the Reviewer and apologize for these mistakes. All modifications were done in the revised version of the manuscript.

  • The conclusion needs to be rewritten, in order to highlight the main aim of the study, the results obtained and the clinical relevance of performing this study.

The conclusion was rewritten accordingly with the Reviewer’s comments.

Round 2

Reviewer 1 Report

The authors have responded adequately to the suggestions, and the work is in an even better form as a result. From my perspective, there are no other remarks to make about this work, which is now complete. I recommend the manuscript for publication.